# Work stress and burnout among active correctional officers in Puerto Rico: A cross-sectional study

**Lisyaima Laureano-Morales<sup></sup>, Nashaly Saldaña-Santiago<sup></sup>, Nitza Malave-Velez, Joshua Quiles-Aponte, Sherrilyz Travieso-Perez, Yaritza Diaz-Algorri** ◉ *, **Alexis Vera**

Master of Public Health Program, San Juan Bautista School of Medicine, Caguas, Puerto Rico, United States of America

☯ These authors contributed equally to this work.
* ydiaz@sanjuanbautista.edu

**Data Availability Statement:** All relevant data are within the manuscript and its Supporting information files.

## Abstract

### Introduction

Correctional officers (COs) are exposed to emotional and physical harm by the nature of their work. Operational stress can lead to burnout and influence absences and COs work performance.

### Objectives

This study aimed to evaluate the association between work-related stress and burnout adjusted by potential confounding variables (age, sex, correctional facility, type of correctional facility, distance to work, and absenteeism).

### Methods

The sample of this cross-sectional study was made up of 799 prison officials. The self-administered questionnaire consisted of four instruments: demographic data, Health and Job Performance Questionnaire, Police Operational Stress Questionnaire, and Maslach Burnout Inventory. The questionnaires were completed online and in person.

### Results

A high proportion of COs reported high operational stress and burnout levels. Fatigue was the highest mean value from all stressors, with 5.89. COs reported high levels of emotional exhaustion and depersonalization. They also reported low levels of personal accomplishment. Furthermore, COs with high stress levels are approximately eight times more likely to experience burnout.

### Conclusion

These findings suggest that COs in Puerto Rico exposed to stress are more vulnerable to present burnout. The findings suggest that evidence-based interventions and programs

**Funding:** The author(s) received no specific funding for this work.

**Competing interests:** The authors have declared that no competing interests exist.

should be implemented to help prevent and reduce operational stress and burnout among COs.

## Introduction

Studies have identified that 37% of Correctional Officers (COs) experience work stress and burnout compared to the general population of 19% [1]. COs are the people who oversee guarding, maintaining order and discipline in correctional institutions, protecting people and property, supervising and offering guidance to inmates, and contributing to their rehabilitation process [2]. Prisons present harsh and hazardous working conditions, fast rates of physical and mental fatigue, risk of infectious diseases, irregular shifts, and low financial reward rates, which systematically decrease employees' quality of life and career fulfillment [3].

When work stress becomes chronic, it strongly affects physical and mental health; today, stress is considered a psychosocial risk in the workplace [4]. When responses to chronic occupational stress are inadequate, burnout emerges as an occupational phenomenon, defined as "*a prolonged response to chronic emotional and interpersonal stressors*" [4]. Burnout occurs in response to chronic exposure to work-related stressors and consists of three main dimensions: first, emotional burnout, defined as the expressed feelings of fatigue or chronic stress; second, depersonalization or skepticism, that is, the dimension of interpersonal context around burnout about adverse interactions with (e.g.) supervisors, co-workers, users; and third, a significant reduction in perceptions of professional achievement in the workplace [3]. Excessive workload, insufficient staff, conflict of values, inadequate rewards, and poor work environment increase the risk of burnout [3].

Operational stressors are related to policing specificities, such as the conditions of the correctional facility and traumatic events COs can encounter while working [5]. Work stress can become chronic and profoundly affect the physical and mental health of COs, leading them to develop exhaustion and feelings of depersonalization. The structure of correctional institutions and the relationships between management and COs can cause work stress and a feeling of dissatisfaction [1]. COs are exposed to high mental, physiological, and cognitive requirements [6–8]. This repetitive condition can lead to strain and burnout in the population [6]. Officers working in prison experience high psychological pressure due to increased job demands [7]. It has been reported that COs confront the most violent, antisocial, and problematic elements of society, putting themselves in stressful situations daily [8]. Traumatic events have been reported numerous times, including manifestations linked to stress and extreme exhaustion, known as burnout [8].

Even though a limited number of studies have addressed the association between work-related stress and burnout in COs, none have been conducted among Puerto Rican COs. This study addressed this gap by evaluating the association between work-related stress and burnout adjusted by potential confounding variables (age, sex, correctional facility, type of correctional facility (maximum, minimum, and median), distance to work, and absenteeism). By better understanding these exposures and outcomes, public policy and tailored interventions can be created to improve the health and assess the needs of COs in Puerto Rico.

## Methods

### Sample

The study population included 3,100 COs members of the Alianza Correccional Unida (ACU) syndicate, with a sample size of n = 799 COs. The participant flowchart is included in Fig 1.

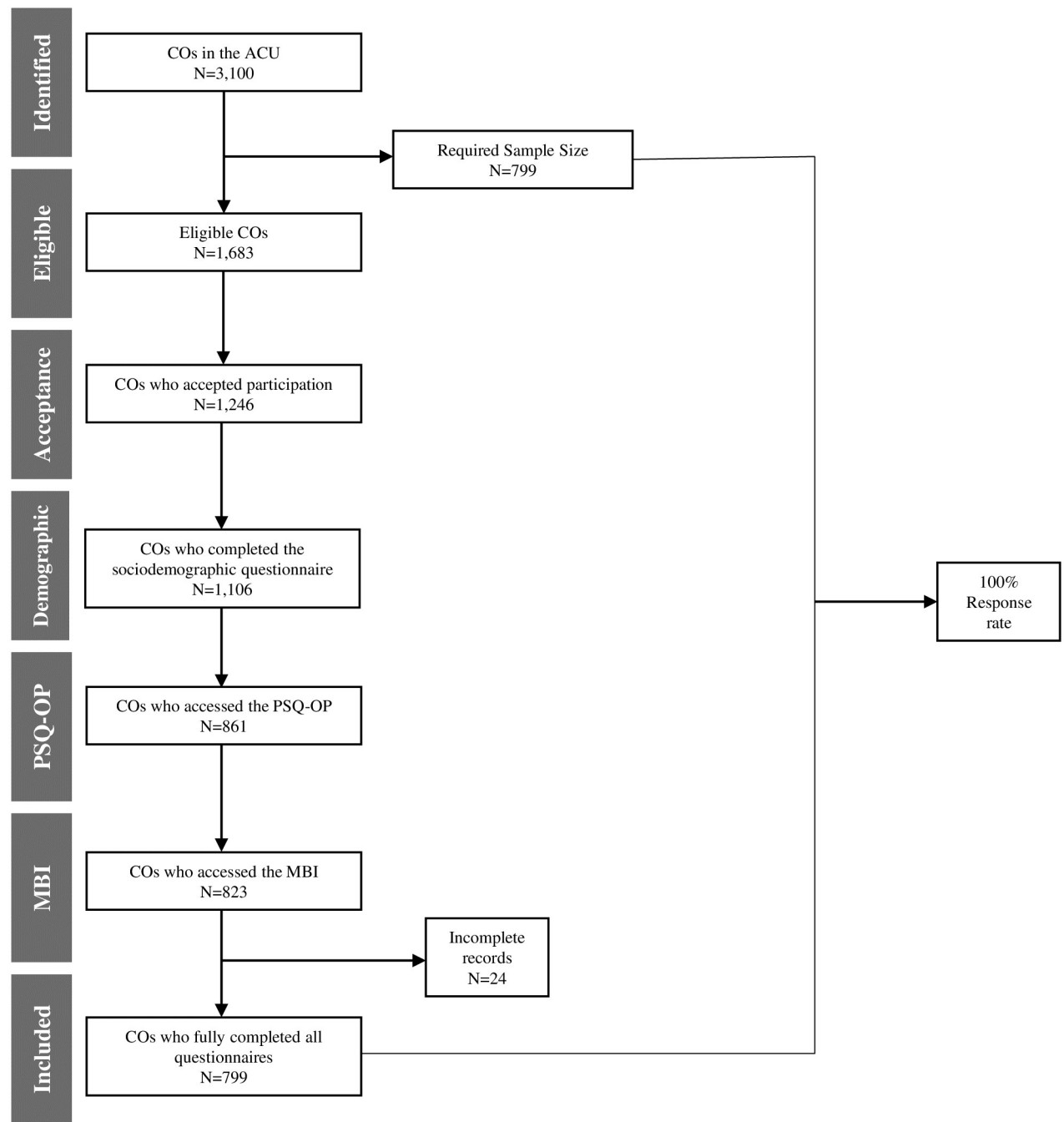

**Fig 1. Flowchart of participant recruitment.**

We calculated the sample size for this study as a convenience sample utilizing EpiInfo with a two-sided confidence level of 95% and a power of 80.0%. The ratio of unexposed (COs not exposed to work stress): to exposed (CO's that were exposed to stress) was 0.7 and calculated based on the data obtained by the literature review. A study by Hernández and Castro in 2013 reported that 27.2% (n = 191 COs) of the participants in Zuera, Spain, had high levels of burnout without stress [9]. We used 27.2% as the outcome for the unexposed group. In Quebec,

Canada, 37% of COs have both stress and burnout [1]. This percentage was utilized as the outcome for the exposed group.

The inclusion/exclusion criteria were: (1) being active in ACU, (2) having served as a correctional officer for more than one year, (3) having more than twenty-one years old, (4) wanting to be part of the study and signing the consent form.

## Procedure, design, and ethics

The study was performed with a cross-sectional design, in collaboration with ACU. It was promoted through email, social media, video capsules, and flyers in collaboration with ACU and its delegates. The participants accessed the questionnaire through the link in the promotions and the QR Code. The data collection stage was held from December 2022 to February 2023 through the participants' self-reporting electronically using REDCap and on paper. The link was available for three months for the electronic self-report, and the self-report on paper was carried out during visits to Penitentiary Institutions in collaboration with ACU.

The anonymity of the participants was protected, and the fact that their participation would be used for research purposes was declared. The study design methodology was evaluated and approved by the Institutional IRB of San Juan Bautista School of Medicine (EMSJIRB-11-2022). The participants required an informed consent statement to start completing the questionnaires.

## Description of questionnaires

The questionnaires were administered in Spanish and consisted of three parts. The first part of the demographic questionnaire included the variables: sex, age, civil status, annual income, years of service, correctional complex, level of security, absenteeism, burnout workshop, and distance to work.

The second section included the Police Operational Stress Questionnaire (PSQ-Op) used to assess occupational stress in police officers [10]. PSQ-Op has 20 items evaluated on a 7-point Likert scale ranging from 1 ("no stress") to 7 ("too much stress"). The PSQ-Op has an interpretation of >3.5 high level, 2.0–3.4 moderate level, and <2.0 low level [8]. We used the validated 20-item Puerto Rican PSQ-Op with Cronbach's alpha results of .94 [11]. The third and final section included the Maslach Burnout Inventory—Human Service Survey (MBI—HSS) to measure burnout [12]. The MBI has a validated Spanish version, consisting of a 22-item 6-point scale measured by three factors (Emotional Exhaustion (EE), Depersonalization (DP), and Personal Accomplishment (PA) [12]. To interpret these three factors, there is a scale that refers to the interpretation of the score obtained by participants, for PA: > = 39 high level, 32–38 moderate level, < = 31 low level. DP can be measured as > = 13 high level, 7–12 moderate level, < = 6 low level. We measure EE as follows: > = 27 high level, 17–26 moderate level, < = 16 low level [13].

## Statistical analysis

We used descriptive statistics to characterize the study population regarding socio-demographic variables. For each socio-demographic variable, the mean and standard deviation were calculated for numeric variables and frequency and percentage for categorical variables. The associations between variables were evaluated using chi-square tests for categorical variables. This study analyzed burnout as a dichotomous dependent variable (Yes burnout, No burnout). Yes burnout was considered as an outcome when a score of $\geq 27$ in Emotional Exhaustion (EE) with a $\geq 13$ in Depersonalization (DP) or $\leq 31$ in Personal Accomplishment (PA) [14, 15]. To analyze the associations between operational stress and burnout, the operational stress

variable was dichotomized using the median value of the PSQ total score (5.05) from which the 2 groups were created: PSQ-Op score >5.05 and PSQ-Op ≤5.05. Given the absence of literature on dichotomization and the data distribution, it was concluded that the median was the correct and ethical trend measure for dichotomization to minimize bias in the interpretation. The multivariable analysis was conducted (multiple logistic regression) to identify independent factors associated with the dependent variable burnout. Variables included in the multivariable analysis were relationship status, annual income, years of service, correctional complex, distance from work, type of facility and occupational stress. Age, sex, and absenteeism were kept in all models due to being potential confounder variables. Missing values MBI and PSQ-Op were imputed by the mode value of each instrument. While three missing values for age were imputed by the mean to execute the models of the multivariate analysis. Due to the data distribution being affected values greater than 105 minutes for the distance variable were defiend as outliers and eliminated utilizing the Interquartile Range method. Only 32 values were eliminated, corresponding to 4% of this variable's data. Results of the associations were reported as odds ratios (OR) with 95% confidence intervals (95% CI) and p-values. All statistical analyses were conducted utilizing R version 4.2.3 and Tableau version 2022.4.1 [16, 17].

## Results

A total of 1,683 records from eligible participants were collected for a sample of 799 participants who completed all questionnaires. The minimum required sample was achieved with 799 COs who completed all the questionnaires. The descriptive socio-demographic statistics of the study are found in Table 1. Also, Fig 2 shows how the correctional complexes were distributed.

Using bivariate correlation analysis, Table 1 allowed us to see the measures of association and significance between the demographic variables and Burnout. After examining the association between burnout and demographic variables, no significant association was found in the following variables: years of service, annual income, age, sex, marital status, correctional complex, and security level. A significant association was found between burnout and Absenteeism, with a p-value of 0.003.

### Operational stress and burnout

The mean score value for the PSQ-Op of our study population was 4.93 (±1.93). As shown in Table 2, among COs, the highest mean value of operational stressors corresponded to "fatigue" with 5.89 (SD ±1.62). Also, as shown in Table 3, the PSQ-Op scores show that 87.10% (n = 696) of the COs reported high levels of stress, 9.63% (n = 77) moderate, and 3.25% (n = 26) low levels of stress.

We used the MBI—HSS to measure burnout using the following scale for PA: ≥39 high levels, 32–38 moderate level, ≤31 low levels; DP: ≥13 high levels, 7–12 moderate level, ≤6 low levels, and EE: ≥27 high levels, 17–26 moderate level, ≤16 low levels [11]. Table 4 shows the mean score for the factors of burnout. EE had a mean of 37.11 (N = 799, SD 13.74), with 79.10% reporting high levels of EE. The mean DP score was 15.95 (N = 799, SD 8.11), with 66.83% having high DP. The mean score for PA was 28.45 (N = 799, SD 10.39), with 59.95 having a low PA. Table 5 shows the results of the multiple logistic regression analysis of stress and confounding variables associated with burnout. COs who score more than 5.05 in the PSQ-Op are approximately 8 (OR 8.21, 95% CI 5.45–11.74) times more likely to experience burnout when adjusting for potential confounder variables sex, age, and absenteeism. Even though sex and age were not confounders in our study, they were included in the models for being traditional confounders in the literature.

**Table 1. Descriptive sociodemographic characteristics of correctional officers and correlations between burnout and sociodemographic characteristics.**

| Variable | Description | Frequency | % or SD | p-value |
|---|---|---|---|---|
| Age | Mean: 43.86 | 796 | ±9.12 | 0.7647 |
| Sex at birth | Male | 657 | 82.23 | 0.6022 |
|  | Female | 142 | 17.77 |  |
| Marital Status | Single | 210 | 26.28 | 0.5596 |
|  | Divorce | 62 | 7.76 |  |
|  | Widowed | 7 | 0.87 |  |
|  | Married | 359 | 44.93 |  |
|  | Living Together | 161 | 20.15 |  |
| Years of service | 1 to 15 | 294 | 36.80 | 0.321 |
|  | 16 or more | 505 | 63.20 |  |
| Annual income | 0 to $30,000 | 393 | 49.19 | 0.709 |
|  | $30,001 or more | 406 | 50.81 |  |
| Correctional Complex | Northern Region | 450 | 56.32 | 0.2918 |
|  | Southern Region | 349 | 43.68 |  |
| Security level at correctional institution | Minimum | 202 | 25.28 | 0.3157 |
|  | Medium | 294 | 36.79 |  |
|  | Maximum | 303 | 37.92 |  |
| In the last 6 months, have you been absent from your duties as Correctional Officer? | Yes | 544 | 68.0 | 0.003* |
|  | No | 255 | 31.9 |  |
| In the last 6 months, have you received any workshop or training on burnout? | Yes | 20 | 2.50 | 0.5143 |
|  | No | 779 | 97.4 |  |
| Distance from home to work (minutes) | Mean: 38.12 | 767 | ±20.80 | 0.5011 |

P values are significant at ≤ 0.05 and were obtained utilizing Chi-square

*Association is significant

## Discussion

Our study responds to the call made by the scientific community to carry out studies on the prevention of burnout and work stress management [1]. This study evaluated the association between work stress and burnout in active Correctional Officers. To our knowledge, this is the first research addressing this association in COs of Puerto Rico adjusted by potential confounding variables (age, sex, correctional facility, type of correctional facility (maximum, minimum, and median), distance to work, and absenteeism).

After evaluating the demographic characteristics between burnout and COs in Puerto Rico, we found absenteeism and work stress related to the rise in burnout. The levels of emotional

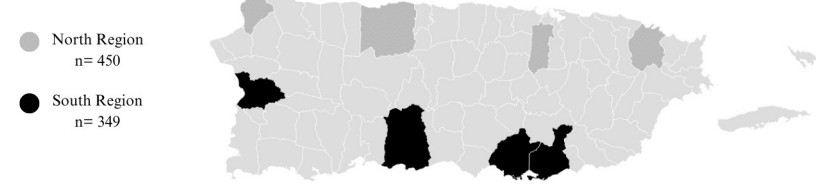

**Fig 2. Distribution of correctional complexes by region.**

**Table 2. Operational stressors among correctional officers.**

| PSQ-Op Item | Mean Score | SD |
|---|---|---|
| Shift work | 5.30 | 1.70 |
| Working at night alone | 4.89 | 2.21 |
| Over time demands | 5.51 | 1.81 |
| Risk of being injured on the job | 5.71 | 1.74 |
| Work related activities on days off (e.g., court, community events) | 4.24 | 2.23 |
| Traumatic events (e.g., MVA, domestics, death, injury) | 4.47 | 2.28 |
| Managing your social life outside of work | 3.95 | 2.02 |
| Not enough time available to spend with friends and family | 5.43 | 1.81 |
| Paperwork | 4.44 | 1.88 |
| Eating healthy at work | 5.09 | 2.08 |
| Finding time to stay in good physical condition | 5.21 | 1.96 |
| Fatigue (e.g., shift work, over-time) | 5.88 | 1.62 |
| Occupation-related health issues (e.g., back pain) | 5.72 | 1.76 |
| Lack of understanding from family and friends about your work | 4.77 | 2.04 |
| Making friends outside the job | 3.99 | 2.15 |
| Upholding a "higher image" in public | 4.38 | 2.15 |
| Negative comments from the public | 4.69 | 2.15 |
| Limitations to your social life (e.g., who your friends are, where you socialize) | 4.33 | 2.13 |
| Feeling like you are always on the job | 5.51 | 1.83 |
| Friends / family feel the effects of the stigma associated with your job | 5.10 | 1.96 |

exhaustion and depersonalization found in this study were higher among COs from Puerto Rico than those reported in a similar study with a Hispanic population of COs from Colombia [3]. Consistent with these findings, the levels of personal accomplishment were lower for COs in Puerto Rico than those reported for Colombian COs [3]. The prevalence of burnout among COs in Puerto Rico was 70.96% which is higher than that reported in the literature for the general population (19%) [1]. When we compare the burnout factors of COs in Puerto Rico with those of American nurses (EE = 21.2, DP = 5.4, PA = 39.1), the COs in this study have a higher percentage of burnout [18]. These findings suggest that COs in Puerto Rico and other places are more vulnerable to burnout than other occupations. A statistical significance (p-value = 0.003) was found between burnout and absenteeism in the last six months, which indicates that officers who were absent in the last six months report having burnout. Another study done on officers suggests that absenteeism is used as a coping mechanism for burnout [19], which could explain the association between burnout and absenteeism.

The PSQ-Op measures 20 factors to determine stress levels; the highest mean score reported stress factors were: fatigue (5.89), occupational-related health issues (5.72), risk of being injured on the job (5.71), over-time demand (5.51), limitations to social life (5.51) and not enough time available to spend with friends and family (5.43). Those stressors are found in the literature as some of the principal stressors in other studies like in Paleksic, where fatigue (4.4),

**Table 3. Operational stress level among correctional officers.**

| | Low | | Moderate | | High | |
|---|---|---|---|---|---|---|
| | Frequency | Percent | Frequency | Percent | Frequency | Percent |
| PSQ-Op Score | 26 | 3.25% | 77 | 9.63% | 696 | 87.10% |

**Table 4. Mean scores for burnout factors among correctional officers in Puerto Rico.**

| Factors | Low | | Moderate | | High | |
|---|---|---|---|---|---|---|
| | Frequency | Percent | Frequency | Percent | Frequency | Percent |
| Emotional Exhaustion | 87 | 10.89% | 80 | 10.01% | 632 | 79.10% |
| Depersonalization | 126 | 15.77% | 139 | 17.40% | 534 | 66.83% |
| Personal Accomplishment | 479 | 59.95% | 174 | 21.78% | 146 | 18.27% |

shift work (3.4), over time demands (3.6) have the highest mean values. In another study by Carleton, fatigue (3.91) and occupation-related health issues (3.72) were part of the principal stressors [20]. Another important finding is that the PSQ-Op, the mean score value obtained in our study, 4.93, was higher than the mean scores reported by two other studies in Canada, where the means were 3.06 and 3.31 [20, 21]. Even though the limited literature on a population culturally similar to the COs of Puerto Rico is important to highlight, findings may suggest that COs worldwide express similar stress levels and are affected by the same stressors.

When comparing our main objective with the literature, to our knowledge, no previous study in the United States has utilized the PSQ-Op and MBI to evaluate the association between stress and burnout in COs. However, a similar study in China evaluated this association utilizing the MBI for burnout and the Effort-Reward Imbalance questionnaire for work-stress. This study found an association between work-related stress and burnout. This indicates that COs who report stress are more likely to have burnout than those who do not report stress [22]. Although China and Puerto Rico are culturally different, both groups of COs have similar working conditions where they have long shifts and poor pay while having a high-risk work setting.

The high levels of work-stress and burnout reported among COs in Puerto Rico could have an impact on public health. Since it has been reported that high levels of burnout are associated with negatively affecting ones mental and physical health causing depression and anxiety [23]. Therefore, further research on managing and preventing operational stress and burnout is essential. The literature suggests that implementing evidence-based practices can decrease the levels of stress and burnout in correctional officers and other human services professionals [24]. Interventions such as mindfulness, peer-mentoring programs, and cognitive behavioral therapy have shown positive results in reducing high stress levels and burnout [25–27]. For the proposed interventions to be more effective, the American Psychological Association recommends tailoring such interventions to gender, gender identity, culture, ethnicity, race, age, family context, religious beliefs, and sexual orientation. The World Health Organization has also suggested that such interventions and programs to help address burnout should be implemented by organizations like the ACU and the Department of Correction and Rehabilitation of Puerto Rico [28]. Health organizations suggest that employers should ensure their health policies are focused on preventing and managing burnout for their employees [28]. We suggest creating public-private alliances to help with the feasibility of implementing these

**Table 5. Association between operational stress and burnout adjusted by potential confounding variables.**

| Models | n | OR | CI (95%) |
|---|---|---|---|
| Operational stress and burnout | 799 | 8.04 | 5.55,11.90 |
| Operational stress and burnout adjusted by absenteeism | 799 | 7.90 | 5.44, 11.70 |
| Operational stress and burnout adjusted by absenteeism, sex, and age | 799 | 8.21 | 5.45,11.74 |

interventions. In the future, the authors recommend that researchers evaluate the impact of the manifestation of high stress levels and burnout on the physical health of COs. Finally, this methodology can be replicated in other public safety personnel (police, paramedics, firefighters, and dispatchers) [20].

## Strengths and limitations

The current study had several important strengths. First, the sample size (n = 799) was large enough to represent the study population. Second, the questionnaires utilized have been psychometrically validated and are used most for the domain under study. Third, the results may directly or indirectly benefit the COs population in Puerto Rico.

Although several strengths were found, this study also had limitations that may be of direction for future studies. First, this study identified self-reported and recall biases that affected the data quality of the World Health Organization Health and Performance Questionnaire (HPQ). Based on these biases, the data collected in the HPQ could not be analyzed. Another limitation identified was that COs rotate in all 3 security levels. This restricted our ability to assess associations between each security level and burnout. Finally, we did not assess the specific type of job executed within our study population. Therefore, assessing these limitations could benefit future studies with COs.

## Conclusion

This study's results indicate a significant association between work-related stress and burnout in COs of Puerto Rico. According to this, COs exposed to work stress are eight times more likely to present burnout, which could influence the high prevalence of burnout reported. The findings suggest that evidence-based interventions and programs should be implemented to help prevent and reduce operational stress and burnout among COs.

## Supporting information

**S1 Dataset.**
(XLSX)

**S1 Checklist. STROBE statement—Checklist of items that should be included in reports of observational studies.**
(DOCX)

## Acknowledgments

The authors thank Jessica Martínez, president of the United Correctional Alliance of Puerto Rico, for the main collaboration. They also thank Dr. Alexander Delgado Ramos, Dr. Abner Vélez Vega, and Dr. Rita Vélez Alvarado for translating and facilitating the PSQ-Op. They also thank the ACU delegates, Héctor R. Travieso Miranda, Juan B. González Rivera, Sheila Santiago Rivera, Jonathan Blass Zapata, Omar Oliver Hernández, Genaro Collazo Rosario, and Byron Aquino Rodríguez, who contributed to the promotion and visits for data collection. In addition, the authors want to thank Victor M Hernandez Robles for his collaboration in the literature review, data collection stage, and thesis presentation. Finally, the authors thank Dr. Yocasta Brugal, president of the San Juan Bautista School of Medicine (SJBSM) for institutional support and Dr. Estela S. Estapé, director of the SJBSM Research Center, for her review and editing of the manuscript.

## Author Contributions

**Conceptualization:** Lisyaima Laureano-Morales, Nashaly Saldaña-Santiago, Nitza Malave-Velez, Joshua Quiles-Aponte, Sherrilyz Travieso-Perez.

**Formal analysis:** Lisyaima Laureano-Morales, Nashaly Saldaña-Santiago, Nitza Malave-Velez, Joshua Quiles-Aponte, Sherrilyz Travieso-Perez.

**Funding acquisition:** Nashaly Saldaña-Santiago, Joshua Quiles-Aponte, Yaritza Diaz-Algorri.

**Investigation:** Lisyaima Laureano-Morales, Nashaly Saldaña-Santiago, Nitza Malave-Velez, Joshua Quiles-Aponte, Sherrilyz Travieso-Perez.

**Methodology:** Lisyaima Laureano-Morales, Nashaly Saldaña-Santiago, Nitza Malave-Velez, Joshua Quiles-Aponte, Sherrilyz Travieso-Perez.

**Project administration:** Lisyaima Laureano-Morales, Nashaly Saldaña-Santiago.

**Resources:** Lisyaima Laureano-Morales, Nashaly Saldaña-Santiago, Nitza Malave-Velez, Joshua Quiles-Aponte.

**Software:** Yaritza Diaz-Algorri, Alexis Vera.

**Supervision:** Yaritza Diaz-Algorri, Alexis Vera.

**Validation:** Lisyaima Laureano-Morales, Nashaly Saldaña-Santiago, Yaritza Diaz-Algorri, Alexis Vera.

**Visualization:** Lisyaima Laureano-Morales, Nashaly Saldaña-Santiago, Nitza Malave-Velez, Joshua Quiles-Aponte, Sherrilyz Travieso-Perez.

**Writing – original draft:** Lisyaima Laureano-Morales, Nashaly Saldaña-Santiago, Nitza Malave-Velez, Joshua Quiles-Aponte, Sherrilyz Travieso-Perez.

**Writing – review & editing:** Lisyaima Laureano-Morales, Nashaly Saldaña-Santiago, Yaritza Diaz-Algorri, Alexis Vera.

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
