## [Decision Letter · Decision Letter 0]

25 Oct 2023

PONE-D-23-15710“Association between work stress and burnout among active correctional officers in Puerto Rico: a cross-sectional study”PLOS ONE

Dear Dr. Diaz-Algorri,

Thank you for submitting your manuscript to PLOS ONE. After careful consideration, we feel that it has merit but does not fully meet PLOS ONE’s publication criteria as it currently stands. Therefore, we invite you to submit a revised version of the manuscript that addresses the points raised during the review process.

We look forward to receiving your revised manuscript.

Academic Editor Comments:

We appreciate the manuscript you submitted. After a thorough evaluation by our esteemed reviewers, we are delighted to inform you that your manuscript requires revisions.

The reviewers found your work to be promising, and with the suggested revisions, we believe it will significantly enhance the overall quality and impact of the publication.

Information regarding BURNOUT could be found in article: Who Cares What the Doctor Feels: The Responsibility of Health Politics for Burnout in the Pandemic

link : https://www.mdpi.com/2227-9032/9/11/1550

Kindly review and evaluate the requested work to determine whether they are relevant and should be cited.

Once you have completed the revisions, please submit the revised version of your manuscript through our online submission system. Include both a clean version and a tracked changes version to aid our review process.

We eagerly anticipate the enhanced version of your manuscript and your continued collaboration with our journal.

Should you have any questions or require any clarification during the revision process, please do not hesitate to contact us.

Sincerely,

Reviewers' comments:

Reviewer's Responses to Questions

**Comments to the Author**

1. Is the manuscript technically sound, and do the data support the conclusions?

Reviewer #1: Yes

Reviewer #2: Yes

Reviewer #3: Yes

2. Has the statistical analysis been performed appropriately and rigorously? 

Reviewer #1: Yes

Reviewer #2: Yes

Reviewer #3: Yes

3. Have the authors made all data underlying the findings in their manuscript fully available?

Reviewer #1: No

Reviewer #2: Yes

Reviewer #3: Yes

4. Is the manuscript presented in an intelligible fashion and written in standard English?

Reviewer #1: Yes

Reviewer #2: No

Reviewer #3: Yes

5. Review Comments to the Author

Reviewer #1: "Association between work stress and burnout among active correctional officers in Puerto Rico: a cross-sectional study" aims to investigate the association between work stress and burnout among active correctional officers in Puerto Rico. The study is important as it sheds light on the prevalence of burnout among correctional officers and the factors that contribute to it. Overall, the study is important as it contributes to the understanding of burnout among correctional officers and can inform the development of interventions to improve their well-being.

That being said, the study used a convenience sample, which may introduce bias and limit the generalizability of the findings. It would have been preferable to use a random sampling method to ensure a more representative sample of the population of correctional officers in Puerto Rico. Furthermore, the ratio of unexposed (COs not exposed to work stress) to exposed (COs exposed to stress) was calculated based on data obtained from the literature review. While this approach can provide some insights, it may not accurately reflect the specific context of Puerto Rico and the population of correctional officers being studied. It would have been more ideal to collect data directly from the study participants to determine the ratio of unexposed to exposed.

The inclusion/exclusion criterion of the study is that they may limit the generalizability of the findings. The criteria included being active in the Alianza Correccional Unida (ACU), having served as a correctional officer for more than one year, being over 21 years old, and wanting to be part of the study and signing the consent form. While these criteria may have been necessary for the specific population being studied, they may not accurately reflect the broader population of correctional officers in Puerto Rico or other regions. For example, the criteria of being a member of the ACU may exclude correctional officers who are not part of the union, and the requirement of having served for more than one year may exclude newer officers who may have different experiences with work stress and burnout. Additionally, the criteria of wanting to be part of the study and signing the consent form may introduce self-selection bias, as those who are more interested in the topic may be more likely to participate.

The mentioned questionnaire administration is the lack of consideration for cultural and linguistic diversity. The study mentions that the questionnaires were administered in Spanish, which suggests that participants were required to have proficiency in the Spanish language. This language restriction may unintentionally exclude individuals who are not fluent in Spanish, potentially limiting the diversity and representativeness of the sample.

Additionally, the use of questionnaires that have been validated in a specific language (Spanish in this case) raises concerns about the validity and equivalency of the translated versions. Differences in language and cultural nuances can affect the interpretation and understanding of the questionnaire items, potentially affecting the accuracy of the data collected.

To enhance the inclusivity of the study and ensure broader applicability of the findings, it would have been beneficial to provide translated versions of the questionnaires or consider administering them in multiple languages to accommodate the diverse backgrounds of potential participants.

The study did not explore the potential causes of the high levels of burnout among correctional officers in Puerto Rico. While the study found a high prevalence of burnout among correctional officers in Puerto Rico and identified work stress and absenteeism as factors related to burnout, it did not investigate the underlying causes of these issues. Understanding the root causes of work stress and absenteeism could provide insights into potential interventions to reduce burnout among correctional officers. Additionally, the study did not explore the potential impact of burnout on the physical health of correctional officers, which could be an important area for future research.

Furthermore, while the study identified potential interventions to reduce stress and burnout among correctional officers, such as mindfulness, peer-mentoring programs, and cognitive behavioral therapy, it did not explore the feasibility or effectiveness of these interventions in the specific context of Puerto Rico. It is important to consider cultural and contextual factors when implementing interventions to ensure their effectiveness and relevance.

Overall, while the study provides important insights into the high levels of burnout among correctional officers in Puerto Rico, it did not explore the underlying causes of work stress and absenteeism or the potential impact of burnout on physical health. Additionally, the study did not explore the feasibility or effectiveness of potential interventions in the specific context of Puerto Rico.

Reviewer #2: Minor comments:

In line 62, the following sentence “When work stress becomes chronic, it strongly affects physical and mental health; today, 63 stress is considered a psychosocial risk in the workplace” should have a citation.

General Comments:

• Citations should be “[9].”, not “.[9]”

• Include the response rate and why 884 were excluded from the analysis. You could maybe include a sample flowchart.

• Please define what municipalities include the northern region and southern region, it is unclear. You could maybe include a map.

• Please include if any pilot testing was conducted for the survey instruments.

• The authors may want to address the potential public health impacts of their findings, since burnout among COs can also affect broader societal results.

• In the discussion, you should add, what steps could corrections facilities take to improve support for officers and address occupational burnout?

• Consider external factors that might influence burnout rates among COs in Puerto Rico. Are their political, economic, lack of personal, power outages, or sociocultural factors that could be exacerbating the issue?

Reviewer #3: The manuscript is well written. Suggestions were made to the discussion section where authors can discuss more in depth comparisons with other studies and implications of current results. Additionally, a recommendation about cultural differences between countries compared should be provided.

We suggest eliminating the words "association between" from the tittle and using the following: "Work stress and burnout among active correctional officers in Puerto Rico: a cross-sectional study"

6. PLOS authors have the option to publish the peer review history of their article (what does this mean?). If published, this will include your full peer review and any attached files.

Reviewer #1: No

Reviewer #2: **Yes: **Laura T. Cabrera-Rivera

Reviewer #3: **Yes: **Nancy R. Cardona Cordero

---

## [Author Response · Author response to Decision Letter 0]

21 Dec 2023

THE RESPONSE OF REVIEWERS WAS ATTACHED AS A FILE DOCUMENT.

---

## [Decision Letter · Decision Letter 1]

6 Mar 2024

PONE-D-23-15710R1“Work stress and burnout among active correctional officers in Puerto Rico: a cross-sectional study”PLOS ONE

Dear Dr. Diaz-Algorri,

Thank you for submitting your manuscript to PLOS ONE. After careful consideration, we feel that it has merit but does not fully meet PLOS ONE’s publication criteria as it currently stands. Therefore, we invite you to submit a revised version of the manuscript that addresses the points raised during the review process.

 Your paper has been re-reviewed. While the comments of all the referees are quite positive, there is a short set of issues still needing your attention. Please refer to the comments provided by our Reviewer # 2 regarding the statistical treatment (e.g., outliers and possible impacts of analytic decisions on the study outcomes). Please try to provide a very thoughtful and reasoned set of responses in these regards, in order to ask the reviewer for a prompt reconsideration of their editorial suggestion.

We look forward to receiving your revised manuscript.

Kind regards,

Sergio A. Useche, Ph.D.

Academic Editor

PLOS ONE

Journal Requirements:

Reviewers' comments:

Reviewer's Responses to Questions

**Comments to the Author**

1. If the authors have adequately addressed your comments raised in a previous round of review and you feel that this manuscript is now acceptable for publication, you may indicate that here to bypass the “Comments to the Author” section, enter your conflict of interest statement in the “Confidential to Editor” section, and submit your "Accept" recommendation.

Reviewer #1: All comments have been addressed

Reviewer #2: All comments have been addressed

Reviewer #3: All comments have been addressed

2. Is the manuscript technically sound, and do the data support the conclusions?

Reviewer #1: Partly

Reviewer #2: Yes

Reviewer #3: Yes

3. Has the statistical analysis been performed appropriately and rigorously? 

Reviewer #1: Yes

Reviewer #2: Yes

Reviewer #3: Yes

4. Have the authors made all data underlying the findings in their manuscript fully available?

Reviewer #1: Yes

Reviewer #2: Yes

Reviewer #3: No

5. Is the manuscript presented in an intelligible fashion and written in standard English?

Reviewer #1: Yes

Reviewer #2: Yes

Reviewer #3: Yes

6. Review Comments to the Author

Reviewer #1: (No Response)

Reviewer #2: While removing outliers for the distance variable is a reasonable approach to avoid skewing the data distribution, providing more context around how outliers were defined and what proportion of data was removed would allow for better assessment of the impact this preprocessing step had on the analysis results.

Reviewer #3: (No Response)

7. PLOS authors have the option to publish the peer review history of their article (what does this mean?). If published, this will include your full peer review and any attached files.

Reviewer #1: **Yes: **Yohannes Habtegiorgis Abate

Reviewer #2: No

Reviewer #3: No

---

## [Author Response · Author response to Decision Letter 1]

18 Apr 2024

THE RESPONSE OF REVIEWERS WAS ATTACHED AS A FILE DOCUMENT.

---

## [Decision Letter · Decision Letter 2]

21 May 2024

“Work stress and burnout among active correctional officers in Puerto Rico: a cross-sectional study”

PONE-D-23-15710R2

Dear Dr. Diaz-Algorri,

We’re pleased to inform you that your manuscript has been judged scientifically suitable for publication and will be formally accepted for publication once it meets all outstanding technical requirements.

Kind regards,

Sergio A. Useche, Ph.D.

Academic Editor

PLOS ONE

Additional Editor Comments (optional):

Thanks so much for the soundness of your amendments and improvements.

Reviewers' comments:

Reviewer's Responses to Questions

**Comments to the Author**

1. If the authors have adequately addressed your comments raised in a previous round of review and you feel that this manuscript is now acceptable for publication, you may indicate that here to bypass the “Comments to the Author” section, enter your conflict of interest statement in the “Confidential to Editor” section, and submit your "Accept" recommendation.

Reviewer #2: All comments have been addressed

2. Is the manuscript technically sound, and do the data support the conclusions?

Reviewer #2: Yes

3. Has the statistical analysis been performed appropriately and rigorously? 

Reviewer #2: Yes

4. Have the authors made all data underlying the findings in their manuscript fully available?

Reviewer #2: Yes

5. Is the manuscript presented in an intelligible fashion and written in standard English?

Reviewer #2: Yes

6. Review Comments to the Author

Reviewer #2: (No Response)

7. PLOS authors have the option to publish the peer review history of their article (what does this mean?). If published, this will include your full peer review and any attached files.

Reviewer #2: **Yes: **Laura T. Cabrera-Rivera

---

## [Editor Report · Acceptance letter]

22 Aug 2024

PONE-D-23-15710R2 

PLOS ONE

Dear Dr. Diaz-Algorri, 

I'm pleased to inform you that your manuscript has been deemed suitable for publication in PLOS ONE. Congratulations! Your manuscript is now being handed over to our production team.

Kind regards, 

on behalf of

Dr. Sergio A. Useche 

Academic Editor

PLOS ONE